# Identification of Yak’s *TLR4* Alternative Spliceosomes and Bioinformatic Analysis of TLR4 Protein Structure and Function

**DOI:** 10.3390/ani11010032

**Published:** 2020-12-26

**Authors:** Xingdong Wang, Jie Pei, Pengjia Bao, Chunnian Liang, Min Chu, Shaoke Guo, Ping Yan, Xian Guo

**Affiliations:** 1Key Laboratory of Yak Breeding Engineering of Gansu Province, Lanzhou Institute of Husbandry and Pharmaceutical Sciences Chinese Academy of Agricultural Sciences, Lanzhou 730050, China; wxd17339929758@163.com (X.W.); peijie@caas.cn (J.P.); baopengjia@caas.cn (P.B.); chunnian2006@163.com (C.L.); chumin@caas.cn (M.C.); gsk1125@163.com (S.G.); 2Life Science and Engineering College, Northwest University for Nationalities, Lanzhou 730030, China

**Keywords:** Yak, *TLR4* gene, alternative splicing, I-TASSER, PDB database

## Abstract

**Simple Summary:**

In this study, yak’s *TLR4* gene alternative spliceosomes were investigated using PCR amplification and cloning with an aim to improve disease-resistance in yaks and promote efficient utilization of yak’s resources. qRT-PCR was used to evaluate the expression levels of two alternatively spliced *TLR4* transcripts in seven distinct yak tissues. To predict the function of proteins expressed by each *TLR4* spliceosome, TLR4 protein structure and function were analyzed bioinformatically. Besides, two alternative spliceosomes of yak’s *TLR4* gene were also identified, which were in line with predicted variants of the *TLR4* gene in NCBI. These two alternative spliceosomes of the *TLR4* gene were expressed in each tissue; however, the expression levels of these spliceosomes were significantly different in different tissue. We also observed that deletion of exon-2 in *TLR4* affected the function of the corresponding protein. This study will lay a theoretical foundation for future studies on the role of two variants of yak’s *TLR4* gene in disease resistance. Besides, data from this study could be analyzed further to explore the molecular mechanism associated with disease-resistance in the yak.

**Abstract:**

In this study, the yak’s *TLR4* gene alternative spliceosomes were investigated using PCR amplification and cloning to improve disease-resistance in yak and promote efficient utilization of yak’s resources. qRT-PCR was used to determine the expression levels of two alternatively spliced transcripts of the *TLR4* gene in seven distinct tissues. To predict the function of proteins expressed by each *TLR4* spliceosome, bioinformatic analysis of yak’s TLR4 protein structure and function was performed, which led to the identification of two alternative spliceosomes of yak’s *TLR4* gene. The TLR4-X1 sequence length was 2526 bp, and it encoded full-length TLR4 protein (841 amino acids). The sequence length of the exon-2 deleted TLR4-X2 sequence was 1926 bp, and it encoded truncated TLR4 protein (641 amino acids). TLR4-X2 sequence was consistent with the predicted sequence of the *TLR4* gene in GenBank. Each tissue showed significantly different expression levels of these two alternative spliceosomes. As per the bioinformatic analysis of the structure and function of TLR4 protein, deletion of exon-2 in the *TLR4* gene resulted in frameshift mutations of the reading frame in the corresponding protein, which altered its ligand-binding and active sites. Besides, biological property such as substrate specificity of truncated TLR4 protein was also altered, leading to altered protein function. This study has laid a theoretical foundation for exploring the role of two variants of the *TLR4* gene in yak’s disease resistance. Besides, this study’s data could be analyzed further to explore the molecular mechanism associated with disease-resistance in the yak.

## 1. Introduction

Yaks are long-haired bovine species and domesticated cattle that are found at plateau located between 3000–5000 meters above sea level. Native to China, 95% of the world’s yak population is distributed around the Qinghai-Tibet plateau in China [1]. Apart from meat and milk, yaks are widely used for labor. Besides, yaks provide livelihood and multiple useful resources, such as hair and skin, for the Qinghai-Tibet plateau’s pastoral areas. Thus, yak is also known as “boat on the plateau” and “omnipotent livestock [2].” However, yaks have a low survival rate due to low immunity [3]. Recently, due to the rising standard of living, the yak market has gradually expanded. Thus, improving the yak’s immunity is crucial for expanding the yak market.

Toll-like receptors (TLRs) are pathogenic pattern recognition receptors (PRPs), which were discovered in 1997 [4,5]. TLRs mediate both innate and acquired immunity and play a crucial role in the immune response [6]. Till now, a total of 13 TLRs have been identified in mammals [7]. TLR4 is one of the earliest discovered and a highly characterized member of the TLR family [8]. It is mainly distributed on the membrane of monocytes, macrophages, NK cells, and adipocytes [9]. TLR4 recognizes highly conserved pathogen-associated molecular patterns (PAMPs) receptors, which are expressed on pathogenic microorganisms [10]. TLR4–PAMP interaction activates intracellular signaling in the early stages of pathogenic invasion to induce the expression crucial immunogenic genes [11]. At the same time, immunogenic genes promote cell synthesis and secretion of cytokines like IFN, IL and TNF. These cytokines transmit upstream signals to downstream and stimulate the natural immune response to kill and elim-inate pathogens, which could further activate the adaptive immune response [12]. A recent study has shown that *TLR4* gene knockout in mice resulted in disorders of visceral adipose tissue and immune cells [13] and that TLR4 was involved in crucial regulatory roles of the innate immune system. Multiple studies have explored the *TLR4* gene in domestic animals [14,15] and poultry [16], but the *TLR4* gene in yak remains unexplored. Unraveling the expression and function of the yak’s *TLR4* gene alternative spliceosomes will be highly significant for future immunogenic studies. This study provides a theoretical basis for exploring the yak’s innate immune system; besides, the data could direct the breeding of high-quality disease-resistant yaks.

Alternative splicing (AS) is the process where different mature mRNA spliced isoforms are produced from a single pre-mRNA through different splicing methods and splice sites [10]. It is a crucial process that regulates gene expression and proteomal diversity [17]. Alternative splicing was first discovered in adenoviruses [18] and subsequently, in mammals [19]. As per the previous studies, alternative splicing plays a crucial role in the generation of receptor diversity, regulation of growth and development, specifically the nervous and immune system; besides, it exerts functional diversity and response sensitivity to each tissue [20]. Multiple pathogenic mutations are co-related to the alternative splicing of genes. Mutations in the conserved sequence of splicing sites, i.e., cis- and trans-acting elements, post alternative splicing, resulted in the physiological abnormalities [21]. Deciphering the correlation between the regulatory mechanism of alternative spliceosomes and diseases could unravel the pathogenic mechanism and facilitate the development of therapeutic agents for multiple immunogenic disorders. 

In this study, the CDS region of the yak’s *TLR4* gene was amplified using primers, which were designed using yak’s *TLR4* gene’ predicted sequences (XM_005891938.1 and XM_014477047.1), obtained using NCBI (https://www.ncbi.nlm.nih.gov/). Apart from identifying alternative spliceosomes, novel alternative spliceosomes of the *TLR4* gene were also explored. qRT-PCR was employed to detect the expression levels of two alternative spliceosomes of the *TLR4* gene in distinct tissues. Furthermore, functional bioinformatics analysis was performed for each spliceosome. This study aimed to reveal the structure, expression characteristics, and functions of each yak’s *TLR4* gene spliceosome, along with structural and functional differences between TLR4 full-length and truncated protein from exon-2 deleted *TLR4* gene. In this study, we investigated the effect of the *TLR4* gene’s alternative splicing on the function of the corresponding TLR4 protein. This study provides primary data for future research on yak’s innate immunity.

## 2. Materials and Methods

### 2.1. Experimental Animals

All experimental procedures and protocols were approved by the Institutional Animal Care and Use Committee of the Lanzhou Institute of Husbandry and Pharmaceutical Science of the Chinese Academy of Agricultural Sciences (Approval No: NKMYD201904; dated: 26 October 2019). In this experiment, three healthy male yaks were procured from Datong Yak Factory, located in Qinghai Province, China. All the three yaks were euthanized, and their heart, liver, spleen, lung, kidney, back muscles, and the subcutaneous adipose tissue of the back were harvested and immediately transported to the Lanzhou Institute of Husbandry and Pharmaceutical Sciences Chinese Academy of Agricultural Sciences, Gansu Province, China on dry ice for further experiments. 

### 2.2. Sample Preparation and cDNA Synthesis

Around 3 g of the tissue samples from each tissue (heart, liver, spleen, lung, kidney, muscles, and fat tissue) were used for RNA extraction. Total RNA was extracted from each tissue using TRIzol reagent (Takara Bio Inc., Dalian, China). To synthesize cDNA from RNA, each RNA sample was diluted to 500 ng/μL and reverse transcribed using the Transcriptor First Strand cDNA Synthesis Kit (Takara Bio Inc., Dalian, China). The synthesized cDNA was stored at −80 °C until further use. The RNA concentration and OD260/280 ratio of the samples were determined using NanoDrop 2000 spectrophotometer (ThermoFisher Scientific, Waltham, MA, USA). RNA concentration and OD260/280 ratio of the samples ranged between 500–5000 ng/μL and 1.9–2.1, respectively. The RNA extraction metrics and tissue-specific metadata were showed in Appendix A. Ribonucleic acid quality was assessed by evaluating the 28S and the 18S rRNA bands on a 1% agarose electrophoretic gel.

### 2.3. Primer Design and Synthesis 

Based on two predicted alternative spliceosome sequences of the *TLR4* gene, obtained using NCBI (https://www.ncbi.nlm.nih.gov/) (XM_00589193.1 and XM_014477041.1), three pairs of primers (P1, P2, P3) were designed using Primer Premier 5.0 software [22] to amplify yak’s *TLR4* gene’s CDS region (Table 1) for the identification of novel spliceosomes and two predicted spliceosomes as mentioned in NCBI. Xi’an Qingke Biotechnology Co., Ltd (Xi’an, China) synthesized the primers.

### 2.4. TLR4 Variants, PCR Amplification, Cloning, and Sequencing

PCR reaction mixture contained 1 μL of forward primer and 1 μL of reverse primer, 12.5 μL of 2 × Taq Master Mix, 1 μL of cDNA from yak’s spleen tissue, and 9.5 μL of ddH_2_O. Initial denaturation was carried out at 94 °C for 5 min, followed by 35 cycles of denaturation at 94 °C for 45 s, annealing at 58 °C for 50 s, extension at 72 °C for 3 min, and a final extension at 72 °C for 10 min. PCR products were detected using 1.2% agar-gel electrophoresis (AGE), visualized, and stored in the automatic digital gel imaging system.

The PCR products were recovered from the agarose-gel using the EasyPure Quick Gel Extraction Kit (Quanshijin Biotechnology Co., Ltd., Beijing, China), as per the manufacturer’s instructions. To prepare recombinant plasmids, pMD19-T vectors (Takara Bio Inc., Dalian, China) were mixed with PCR products and placed in a water bath at 16 °C overnight. These recombinant plasmids were incubated with BL21 competent cells (Takara Bio Inc., Dalian, China) to transform the recombinant plasmids into the BL21 competent cells. 200 μL of bacterial suspension was spread on LB agar (Amp+) plates and incubated at 37 °C for 12 h. Later, single colonies were selected and inoculated into the 5 mL LB broth (Amp+) and incubated with constant shaking at 37 °C for 12 h. Post-incubation, positive bacterial clones were validated using PCR amplification and sequenced by Xi’an Gt Biotechnology Co., Ltd. (Xi’an, China).

### 2.5. Bioinformatics Analysis of TLR4 Protein Structure and Function

The biological function of a protein can be attributed to its 3D-structure [23]. However, the preparation of protein crystals for analyzing the 3D-structure of proteins is a tedious task. In this study, I-TASSER [24] (http://zhanglab.ccmb.med.umich.edu/I-TASSER/), a prediction software, was employed to predict the structure and function of the protein and to construct the 3D-model of the full-length TLR4 and truncated TLR4 protein, encoded by exon-2 deleted *TLR4* gene. Starting from the amino acid sequence, I-TASSER first uses LOMETS [25] (a multi-threading algorithm consisting of several separate threaded programs) to identify homologous structure templates or super secondary structure segments from the PDB library [26] (http://www.pdo.org/pdb/hoine/home.do). To construct the complete structure, it reassembles the LOMETS template, excises continuous aligned fragment structure from the super secondary structure segment, precisely-optimizes the structure model, and determines the free energy conformation. To deduce the biological function of the target protein, the I-TASSER model was matched with the protein obtained from the BioLip library [27], and its functions, ligand binding site, enzyme classification, and other characteristics were inferred from the BioLip template. A phylogenetic tree was constructed based on the amino acid sequences of two alternative spliceosomes of TLR4, as described previously by Xiaoyan Zhang [28].

### 2.6. Quantitative Real-Time PCR (qRT-PCR) Based Analysis of TLR4-X1 and TLR4-X2 mRNA Expression 

Fluorescence quantitative primers (P4, P5) were designed based on the sequences of two alternative *TLR4* spliceosomes obtained by cloning (Table 1). The glyceraldehyde 3-phosphate dehydrogenase gene (*GAPDH*) was used as the reference gene. Semi-quantitative RT-PCR was used to determine the expression levels of *TLR4* transcript in all the yak tissue samples. Fluorescent quantitative PCR reaction system and reaction conditions were as described previously by Wangsheng Zhao [29], and the relative expression of genes was estimated using the 2^−ΔΔCT^ [30] method. Each reaction was repeated three times to obtain the Ct value of each sample. Statistical differences between the expression levels of the *TLR4* splice variants were determined through ANOVA. SPSS version 21.0 [31] was used to perform statistical analyses. *p* < 0.05 or *p* < 0.01 was considered as statistically significant [32].

## 3. Results

### 3.1. Identification of Alternative Splice Variants of the Yak’s TLR4 Gene

A hybrid cDNA library was prepared to elucidate the sequence of *TLR4* alternative spliceosome from spleen tissue of three male yaks (4 years old) using PCR amplification. The resulting PCR products were visualized using 1% agarose gel electrophoresis (Figure 1a,b,d). The target band was extracted from agarose gel and ligated into the PMD19-T vector. Positive clones (Figure 1c) were sequenced on the ABI 3130XL platform. After sequence alignment using MegAlign version 5.10 [33] and sequence splicing using SeqMan [34], the CDS domain sequences of two distinct *TLR4* gene spliceosomes were obtained. The TLR4-X1 sequence length was found to be 2526 bp long (encoding 841 amino acids), identical to the *TLR4* transcript length in cattle, humans, ducks, and other mammals. The TLR4-X2 sequence length was found to be 1926 bp (encoding 641 amino acids). Comparative analysis of two cloned sequences and two predicted sequences of alternative spliceosome using NCBI revealed 100% sequence similarity between these sequences. It indicated that the two variants of the clones were identical to the predicted sequences obtained using NCBI. The results demonstrated at least two alternative spliceosomes of the yak’s *TLR4* gene.

### 3.2. Sequence Analysis of Two Alternative Spliceosomes of the TLR4 Gene

The complete sequence of the CDS region (login number: KF977430.1) of the *TLR4* gene was aligned against its corresponding genome using NCBI’s splice-alignment tool, Splign [35] (https://www.ncbi.nlm.nih.gov/sutils/splign). As per the outcomes, the CDS region of the *TLR4* gene entailed three exons. 1–93 bases constituted the first exonic region, 94–260 bases constituted the second exonic region, and 261–2526 bases constituted the third exonic region. TLR4-X1 sequence was identical to the complete CDS region of the *TLR4* gene. It showed that the TLR4-X1 entailed a complete CDS area. Besides, TLR4-X2 lacked 94–260 bases, indicating that the entire second exonic region of the *TLR4* gene was deleted (Figure 2).

### 3.3. Bioinformatic Analysis

#### 3.3.1. Prediction of the Secondary Structure of Full-Length and Truncated TLR4 Protein 

TLR4 full-length protein contained 205 Helix (H), 88 Strand (S), and 548 Coil (C) (Figure 3). The predicted sequence of the truncated protein encoded by exon-2 deleted *TLR4* transcript is depicted in (Figure 4); it contained 173 Helix (H), 65 Strand (S), and 403 Coil (C).

#### 3.3.2. Tertiary Structure of Predicted Full-Length and Truncated TLR4 Protein

The normalized z-score was used to measure the reliability of the template. Z-score > 1 indicates a high degree of confidence between the queried and the template sequence. As depicted in Table 2 and Table 3, normalized z-scores of all the ten templates were higher than 1. It indicated a high degree of confidence between the queried and template sequence. Among the full-length and truncated TLR4 protein, the highest z-score corresponded to 3j0aA. 

3D-structure of five TLR4 full-length proteins and truncated TLR4 proteins were predicted using the I-TASSER modeling. Generally, predicted protein is based on the structure of the first model (Figure 5a,b). The full-length TLR4 protein structure is depicted in Figure 5a, where blue to red represents the N- to C-terminus. Following parameters were used to construct full-length TLR4 protein: C-Score = −0.09, predicted TM-score = 0.70 ± 0.12, predicted RMSD = 8.7 ± 4.5Ǡ. Figure 5b depicts a truncated TLR4 protein structure, where blue to red represents the N- to C-terminus. Following parameters were used to construct structure of truncated TLR4 protein: C-Score = −0.23, predicted TM-score = 0.68 ± 0.12, predicted RMSD = 8.4 ± 4.5Ǡ. The structural similarity between the target model and the ten closest proteins were ranked as per the TM-score. The highest full-length protein TM-score (3fxiA) corresponded to that of human TLR4-human MD-2-*E. coli* LPS Ra complex. The highest TM-score (3j0aA) for TLR4 truncated protein corresponded to human Toll-like receptor 5.

#### 3.3.3. Functional Prediction of Yak’s TLR4 Full-Length and Truncated Protein 

Functional templates were comprehensively scored using COACH [36] classification based on sequence and structure distribution alignment. The predicted structure and functional template (Table 4) of the full-length TLR4 protein’s ligand-binding site are depicted in Figure 6a. The C-Score or confidence score for predicted ligand binding sites ranged from 0–1. A higher C-Score indicates more reliable predictions. The first functional template (PDB ID: 3fxiA) showed a lower confidence score (C-Score = 0.05) for NAG peptide ligand binding. The ligand-binding site was Lys381, Arg382, Asp405, Asp428 residues. The structure of a ligand-binding site and a functional template (Table 5) of a predicted TLR4-truncated protein is depicted in Figure 6b. The first functional template (PDB ID: 2Z81A) contained a peptide with a lower confidence score for ligand binding (C-Score = 0.05). The binding ligand was 2Z81A00, and the ligand-binding site was Leu276, Lys277, Met278, Ala279, Thr299, Val300 residues. EC (enzyme classification) and active sites of predicted TLR4-truncated protein were identified using CO-FACTOR, whereas the global and local structure of known proteins were compared using the BioLip database and the I-TASSER model. The model’s structure and the active site are depicted in Figure 7a. EC number and PDB ID of the yak’s full-length TLR4 protein in the functional template are depicted in Table 6. This template was predicted to be IgA-specific serine endopeptidase with EC number 3.4.21.72; however, no active site residues were identified. Figure 7b demonstrates the model’s structure, the active site. Table 7 demonstrates the EC number and PDB ID of the truncated TLR4 protein in the functional template. The first model is based on PDB-ID: 2vkzG predicted template. This template was identified to be (Acyl-carrier-protein) S-acetyltransferase with EC numbers 2.3.1.38, and the Oleoyl-(acyl-carrier-protein) hydrolase with EC numbers 3.1.2.14; however, no active site residues were predicted.

#### 3.3.4. Phylogenetic Tree of TLR4 Amino Acid Sequence

As per the amino acid sequence alignment, the amino acid sequence similarity of yak’s TLR4-X1 amino acid sequence to *Bos taurus*, *Ovis aries*, *Sus scrofa*, *Macaca mulatta*, *Pan troglodytes*, *Gorilla gorilla*, *Canis lupus dingo*, and *Nomascus leucogenys* was found to be 98.81%, 94.38%, 81.09%, 75.67%, 76.10%, 76.05%, 76.77%, and 75.63%, respectively. The amino acid sequence similarity of yak’s TLR4-X2 amino acid sequence to *Bos taurus*, *Ovis aries*, *Sus scrofa*, *Macaca mulatta*, *Pan troglodytes*, *Gorilla gorilla*, *Canis lupus dingo*, and *Nomascus leucogenys* was found to be 99.38%, 94.38%, 80.13%, 73.92%, 74.49%, 74.65%, 76.70%, and 74.25%, respectively. Phylogenetic tree of these amino-acid sequences from nine species, including the pig, cattle, and sheep, was constructed using the neighbor-joining method and Mage version 5.0 (Figure 8). The results showed that the yak’s TLR4 had the highest similarity with *Bos taurus*, followed by *Ovis aries*. It indicated that the *TLR4* gene is highly conserved.

### 3.4. Quantification of TLR4-X1 and TLR4-X2 in Different Yak Tissues

qRT-PCR results validated the expression levels of both the variants in the heart, liver, spleen, lung, kidney, muscle, and adipose tissue. The two-alternative spliced variants of *TLR4* were highly expressed in the spleen (Figure 9). As per the statistical analysis, the expression levels of the two alternatives spliceosome in the spleen were significantly higher than the lung (*p* < 0.05) and heart, liver, kidney, muscle, and adipose tissue (*p* < 0.01). As per the expression levels of two variants, i.e., TLR4-X1 and TLR4-X2 in seven tissues including heart, liver, spleen, lung, kidney, muscle, and fat, the differences in expression levels of the two variants in spleen and lung were highly significant (*p* < 0.01). Expression levels of TLR4-X1 and TLR4-X2 in heart, liver, kidney, muscle, and adipose tissue (*p* < 0.05) also differed significantly.

## 4. Discussion

Alternative splicing is a widely occurring phenomenon and closely related to altered protein functions, such as altered amino or carboxy terminus of a protein or addition/deletion of the functional region [37]. Till now, seven alternative splicing methods [38]: variable 3′ splice sites, variable 5′ splice sites, intron retention, mutual exclusion exons, variable initial exon, variable terminal exon, have been proposed. For the first time, in this study, the entire exon-2 was deleted from the *TLR4* transcript to produce TLR4-X2 mRNA. Previous studies have reported two alternatively spliced variants of the *TLR4* gene in duck [10] and mice [39]; three alternatively spliced variants in sheep [10] and pig [40], and four alternatively spliced variants in human [39]. However, alternatively spliced variant of the yak’s *TLR4* gene have not been reported so far. 

In this study, to identify the alternatively spliced variants of yak’s *TLR4* gene, cDNA from yak’s spleen was amplified using PCR. PCR products were visualized on agarose gel electrophoresis. Three bands appeared in one lane (Figure 1a); these bands were recovered from agarose-gel and later cloned. Two bands (929 bp and 750 bp) were consistent with the predicted sequence of alternative spliceosome obtained using NCBI. These three sequences were aligned, and it was found that the starting sequence of the sample in the third lane (300 bp) was identical to the primer’s sequence. Since this sequence might have been mistaken for the upstream primer by the system and later eliminated from the synthesized sequence, the pair of P2 primers (P2) was used to assess the third band (300 bp) on agarose gel electrophoresis. We observed that one lane contained only two bands (756 bp and 600 bp) (Figure 1b), whereas the 300 bp band was due to the similar primer and the 300 bp band sequences. Thus, the target sequence was mistaken for the primer sequence, and the overlapping sequence was eliminated from the synthesized sequence, resulting in three bands. Although two alternatively spliced variants of the yak’s *TLR4* gene were successfully cloned and identified in this study, it is still unclear if the yak’s *TLR4* gene contains other alternative splicing bodies, which demands an in-depth investigation. 

As per the previous studies, the site of alternative splicing might impact the functionality of the encoded protein. Besides, altered peptide sequence also alters the ligand-binding sites, enzyme activity, allosteric regulation, or protein localization [41]. 3D-model of the full-length TLR4 protein and a truncated TLR4 protein encoded by the exon-2 deleted *TLR4* transcript was constructed using the known protein structure and I-TASSER for the functional characterization of the TLR4 protein. The confidence of each model was quantitatively evaluated using the C-Score from the thread template alignment and the structural assembly simulation’s confluence parameters. Mostly, −5~2 C-Score indicates the high quality of the prediction model [42]. TM-score, which measures the structural similarity between two structures, is mostly in the range of 0~1. TM-score > 0.5 indicates similar structures in the same SCOP/CATH folding family [43]. Models with a C-Score > −1.5 and a TM-score > 0.5 have the correct folding structure. Based on the order of amino acids in the primary structure of the protein, TLR4-X1 contained three exons, encoding 841 amino acids. However, TLR4-X2 showed frameshift mutation in the open reading frame due to the deletion of exon-2. The translation was initiated at 434 bp of the CDS region, resulting in one low complexity region encoded by the first exon, four LRR domains were encoded by the second exon, and part of exon-3 was lost. 3D model of the TLR4 full length and truncated protein from exon-2 deleted *TLR4* transcript was predicted. The full-length protein showed C-Score: −0.09, TM-score: 0.72 ± 0.12 and the truncated protein showed C-Score: −0.23, TM-score: 0.68 ± 0.12. However, the C-Scores of both proteins were higher than −1.5. It suggested the high quality of both the TLR4 full-length protein and the truncated protein model. In addition, the template quality of the full-length TLR4 protein was slightly better than the truncated TLR4 protein. TM-scores of the two proteins were compared, which revealed that the TM-scores of both proteins were almost similar and > 0.5. It demonstrated the correct folding structure and high structural similarity between the protein models. The highest TM-score (3fxiA) of the TLR4 full-length protein matched the homologous template structure from the PDB library based on the amino acid sequence, and it corresponded to the human TLR4-human MD-2-*E. coli* LPS Ra complex. The highest TM-score (3j0aA) of the truncated TLR4 protein matched with the TM-score of human Toll-like receptor 5. It suggested that the truncated protein contained frameshift mutation due to the deletion of exon-2 in the *TLR4* transcript, which resulted in the loss of some domains in the resulting protein and altered protein structure. The structure of the protein was closely correlated to its function. Deletion of exon-2 affected the protein function to some extent.

Functional bioinformatics outcome using ligand BioLiP template, which derived target protein binding site and enzyme classification, showed that the full-length TLR4 protein and IgA-specific serine endopeptidase were functionally identical. IgA-specific serine endopeptidase belongs to the IgA-specific serine endopeptidase family (S 6). Family S 6 contains identical enzymes from two gram-negative pathogenic bacteria, *Neisseria gonorrhoeae* and *Haemophilus influenzae*. These peptidases cleave the heavy chains of immunoglobulin A at prolyl bonds in the hinge region [44], resulting in hydrolysis products consisting of the intact antigen-binding Fab and the Fc region of these antibody proteins [45]. 

TLR4 truncated protein, S-acetyltransferase, and Oleoyl-hydrolase are functionally identical. In plastids, acyl-(acyl-carrier-protein) hydrolase (EC 3.1.2.14) releases fatty acids from the end-product of fatty acid synthesis, which forms glycerolipids in the cytoplasm. Two major processes that contribute to the biosynthesis of glycerolipids in plant cells are a) synthesis of fatty acids in plastids and b) incorporation of these fatty acids into glycerolipids in plastids and ER (endoplasmic reticula). Acyl-(acyl-carrier-protein) hydrolase (AH) plays a crucial role in the transportation of fatty acids from plastids to the cytoplasm for glycerolipid synthesis [46]. In this study, the full-length TLR4 protein and truncated TLR4 protein showed altered structure and functions.

The phylogenetic tree constructed from the amino acid sequences of yak and other species established the close evolutionary relationship of yak with other cattle and distant evolutionary relationship with gorillas, in line with the previous studies. Amino acid sequence alignment of yak with other species elucidated 70–99% similarity between amino acid sequences from different species. It indicated that the *TLR4* gene, which plays crucial roles in animals, remained highly conserved throughout the evolutionary process. Previous studies have shown a close correlation between *TLR4* and diseases. For instance, missense mutations in *TLR4* were associated with decreased reactivity to LPS in a minority population [47]. In chicken, *TLR4* polymorphism was associated with susceptibility to *Salmonella* [48]. It further demonstrated the correlation of the *TLR4* gene with immunity. Thus, in-depth research on the *TLR4* gene might improve the host’s resistance to multiple disorders.

TLR4 is primarily expressed by immune cells that participate in host defense, such as monocytes, macrophages, granulocytes, dendritic cells, lymphocytes, epithelial cells, endothelial cells, bone marrow monocytes, and so on [49,50,51]. In this study, qRT-PCR was used to assess the expression levels of two alternatively spliced variants of the *TLR4* gene in seven distinct tissues. The results showed that *TLR4* gene expression levels were significantly different in different tissues, and it was highest in the spleen, followed by the lung. It is in line with previous studies in duck [10], which showed specific *TLR4* gene expression and function. TLR4, a member of the PRR family, is a type I transmembrane glycoprotein. It participates in multiple functions, such as immune and inflammatory response [52]. Independent samples t-test of the TLR4-X1 and TLR4-X2 demonstrated significantly different expression levels of TLR4-X1 and TLR4-X2 in the same tissue. It might be due to different immune responses induced by TLR4-X1 and TLR4-X2 in the yak. *TLR4* expression levels varied between the tissues to meet the functional requirements. In this study, *TLR4* gene expression in yak was found to be strictly regulated. It indicated that the alternative splicing of *TLR4* plays a crucial role in growth and development. The specific function of the two alternatively spliced *TLR4* will be explored in our future studies.

## 5. Conclusions

In this experiment, two alternatively spliced variants of yak’s *TLR4* gene were identified through PCR amplification and cloning techniques. TLR4-X1 and TLR4-X2 were found to be 2526 bp and 1926 bp long, encoding 841 and 641 amino acids, respectively. TLR4-X1 and TLR4-X2 were expressed in seven tissues but with different expression levels, and the highest expression was observed in the spleen.

## Figures and Tables

**Figure 1 animals-11-00032-f001:**
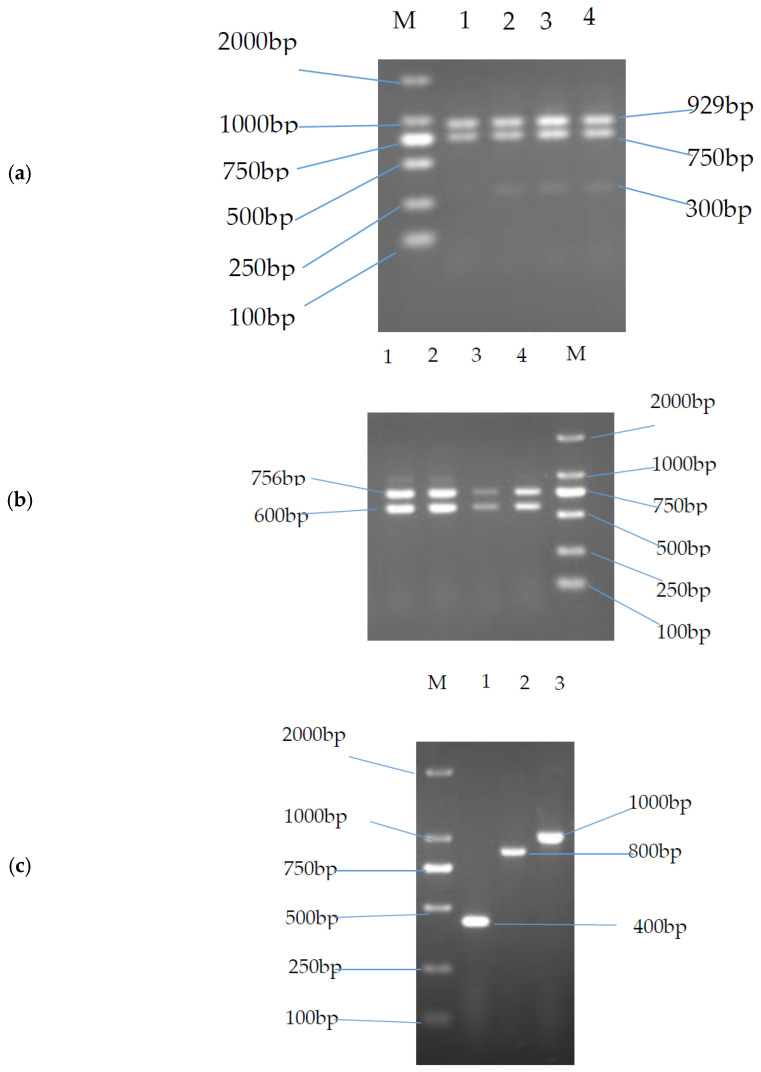
PCR amplification results of *TLR4* gene in Yak. (**a**) Yak *TLR4* alternative spliceosome’s amplification using primer P1. Note: M: DL-2000 NDA (DeoxyriboNucleic Acid) marker; 1–4; RT-PCR product of yak’s *TLR4* alternative spliceosomes; (**b**) Yak *TLR4* alternative spliceosome’s amplification using primer P2. Note: M: DL-2000 NDA marker; 1–4: RT-PCR product of yak’s TLR4 alternative spliceosome; (**c**) Bacterial PCR product of yak’s *TLR4* cloned alternative spliceosomes. Note: M: DL-2000 NDA marker; 1: 300 bp product after cloning of 300 bp product; 2: 750 bp product after cloning of 300 bp product; 3: 929 bp product after cloning of 300 bp product. (**d**) Yak’s *TLR4* alternative spliceosome’s amplification using primer P3. Note: M: DL-2000 NDA marker; 1–4: RT-PCR product of yak’s *TLR4* gene.

**Figure 2 animals-11-00032-f002:**
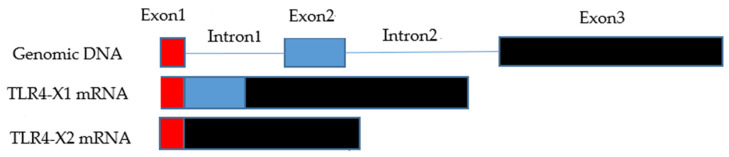
The pattern of alternative splicing in yak’s *TLR4* gene. Note: Red is exon 1, blue is exon 2, and black is exon 3.

**Figure 3 animals-11-00032-f003:**
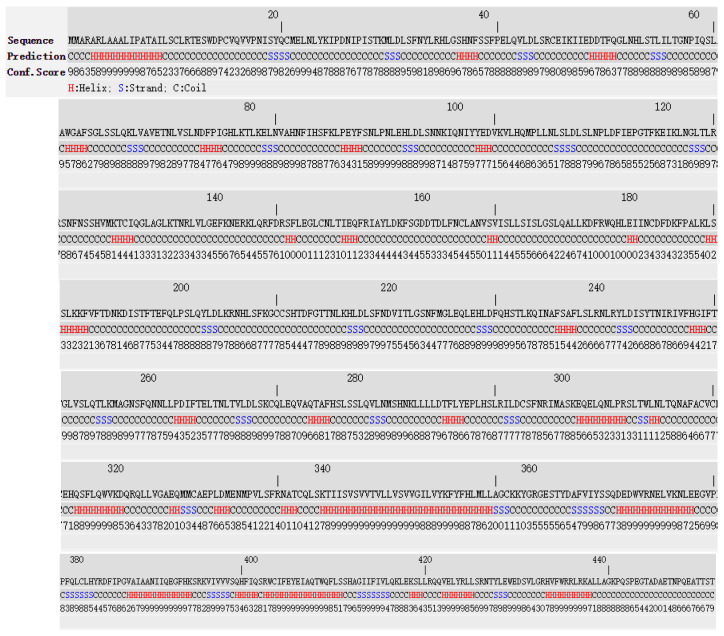
I-TASSER prediction of the secondary structure of yak’s full-length TLR4 protein. Note: H represents Helix, S represents Strand, and C represents coil. The confidence score ranged from 1–10; higher the score, higher is the reliability of the predicted secondary structure.

**Figure 4 animals-11-00032-f004:**
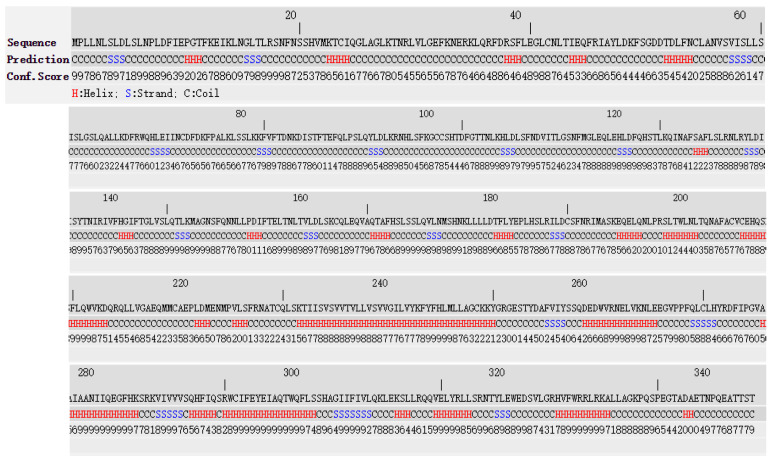
I-TASSER prediction of the secondary structure of the truncated protein encoded by yak’s exon-2 deleted *TLR4* gene.

**Figure 5 animals-11-00032-f005:**
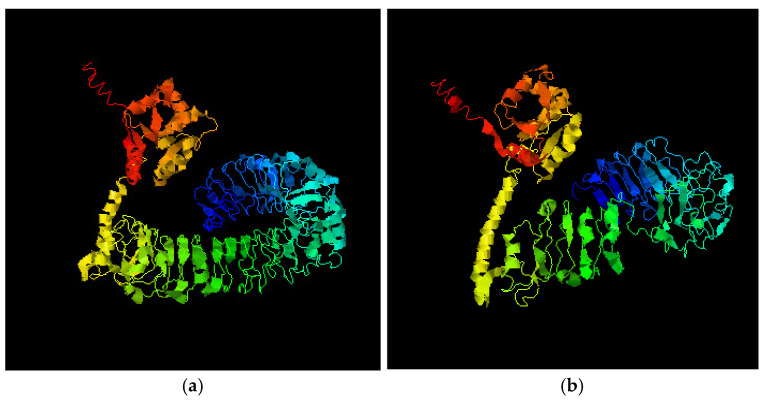
Prediction of TLR4 protein’s tertiary structure: (**a**) Full-length TLR4 protein; (**b**) Truncated TLR4 protein encoded by exon-2 deleted *TLR4* transcript.

**Figure 6 animals-11-00032-f006:**
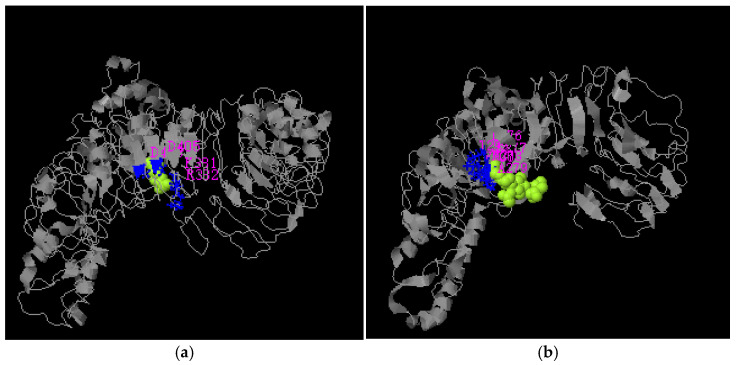
Protein-ligand binding site prediction. (**a**) TLR4 full-length protein-ligand binding site prediction; (**b**) truncated TLR4 protein-ligand binding site prediction.

**Figure 7 animals-11-00032-f007:**
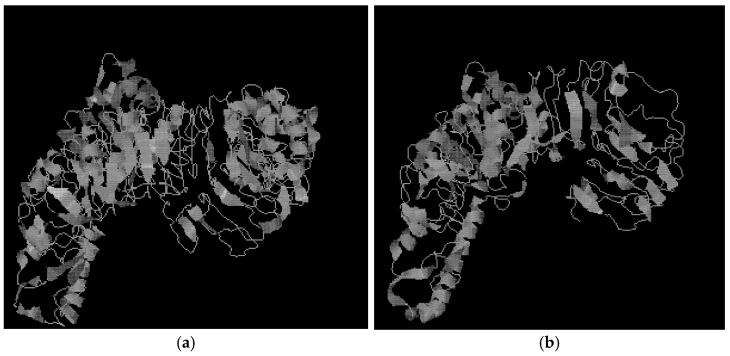
Protease classification and prediction of active sites: (**a**) TLR4 full-length protein; (**b**) truncated TLR4 protein.

**Figure 8 animals-11-00032-f008:**
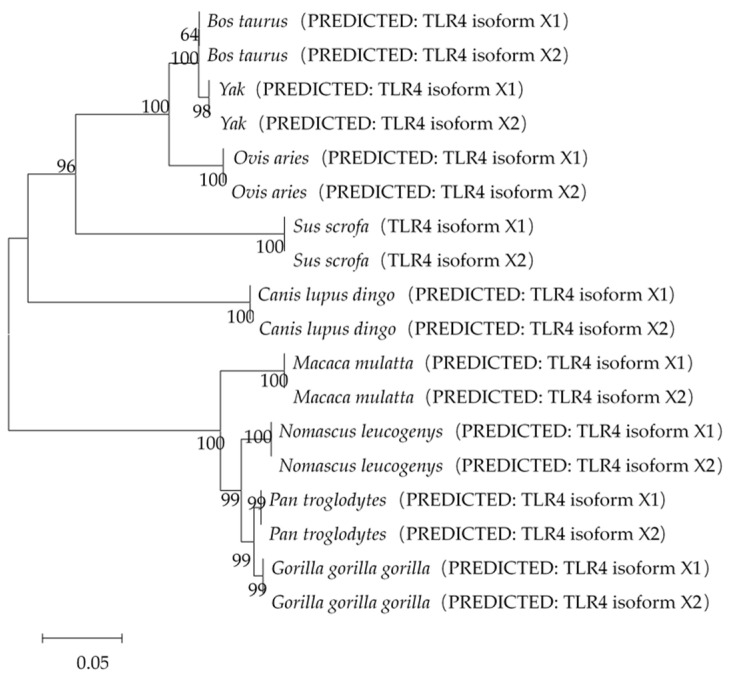
Phylogenetic tree of TLR4 amino acid sequences from nine species: *Bos taurus* (XP_024851214.1; XP_005210643.1), *Ovis aries* (XP_011966604.2; XP_027819693.1), *Sus scrofa* (NP_001106510.2; NP_001280245.1), *Macaca mulatta* (XP_014972446.2; XP_028690038.1), *Pan troglodytes* (XP_009455379.2; XP_009455380.2), *Gorilla gorilla gorilla* (XP_004048563.1; XP_018888525.1), *Canis lupus dingo* (XP_025288404.1; XP_025288405.1), *Nomascus leucogenys* (XP_003264105.4; XP_012364221.2).

**Figure 9 animals-11-00032-f009:**
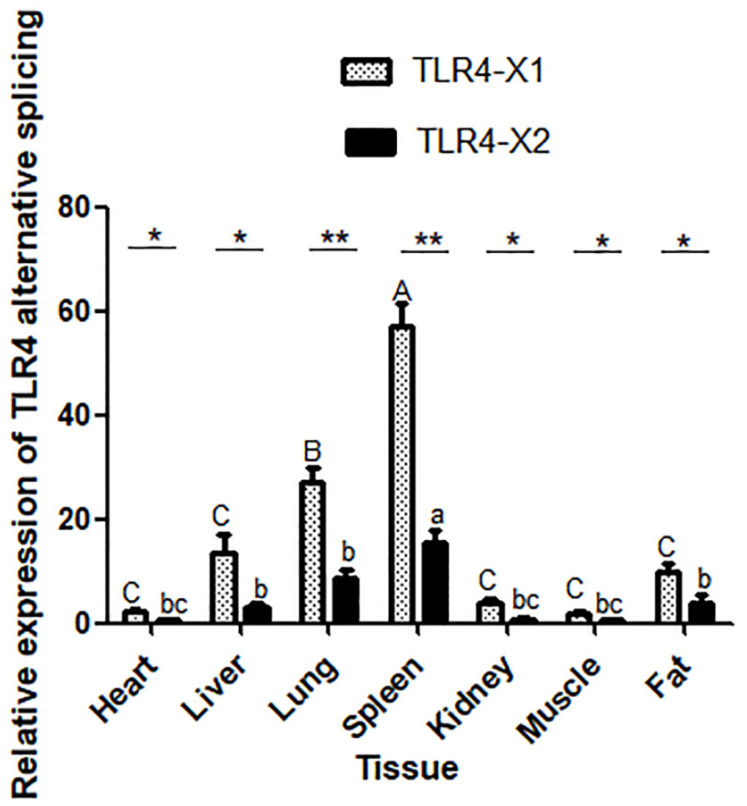
Relative expression of two variants of *TLR4* in different tissues. Note: * indicated that the expressions of TLR4-X1 and TLR4-X2 in the same tissue were significant difference, and ** indicated that the expressions of TLR4-X1 and TLR4-X2 in the same tissue were highly significant difference. A, B and C represent the significance of TLR4-X1 expression in different tissues, while a, b and c represent the significance of TLR4-X2 expression in different tissues.

**Table 1 animals-11-00032-t001:** Primer sequence.

Primer	Primer Sequence (5′–3′)	Product Size (bp)	Annealing Temperature (°C)	Notes
P1	GGTGGCAGAGGAGATAA	929	58	Clone
	CTTCGCAGAGTCAATCC			
P2	CAGACAGAGGGTTATGCT	756	58	Clone
	CTTCGCAGAGTCAATCC			
P3	GCCGTGGAGACAAACCT	2163	58	Clone
	CTGGGAAAGAAGCACATCA			
P4	ACAAAATCCCCGACAAC	92	60	qRT-PCR
	GCTGGAGAAGTTATGGC			
P5	CCTTGCGTACAGATGTG	137	60	qRT-PCR
	GTAAACTTGATAGCCCAGA			
GAPDH	TCACCAGGGCTGCTTTTA	126	60	qRT-PCR
	CTGTGCCGTTGAACTTGC			

**Table 2 animals-11-00032-t002:** Ten thread templates (full-length proteins) used in I-TASSER modeling.

Rank	PDB Hit	Norm. C-Score
1	3j0aA	3.13
2	3j0aA	5.65
3	3j0aA	3.74
4	3j0a	2.35
5	3j0a	1.58
6	3j0aA	5.76
7	3j0a	2.46
8	4g8aA	7.46
9	2z64a	3.93
10	3j0aA	20.10

**Table 3 animals-11-00032-t003:** Ten thread templates used for I-TASSER modeling (deleted second-exon).

Rank	PDB Hit	Norm. C-Score
1	3j0aA	2.65
2	3j0aA	4.69
3	3j0aA	3.06
4	3j0a	2.20
5	3j0a	1.55
6	3j0aA	4.39
7	3j0a	2.85
8	4u71A	5.52
9	2z64a	2.90
10	3j0aA	15.90

**Table 4 animals-11-00032-t004:** TLR4 full-length protein functional template.

Rank	C-Score	Cluster Size	PDB Hit	Lig Name	Ligand Binding Site Residues
1	0.05	5	3fxiA	NAG	381,382,405,428
2	0.02	2	3fxiA	NAG	178,179,207,209
3	0.02	3	2p1nB	CFA	87,108,109,110,132,133,136,156,157,158
4	0.02	3	3a79A	PXS	91,92,97,109,133,136,138,144,157,160,161,162,168,171
5	0.02	3	1ZIWA	1ZIWA07	453,454,455,476,477,478,479,497

**Table 5 animals-11-00032-t005:** Truncated TLR4 protein functional template.

Rank	C-Score	Cluster Size	PDB Hit	Lig Name	Ligand Binding Site Residues
1	0.05	6	2Z81A	2Z81A00	276,277,278,279,299,300
2	0.03	3	3fxiA	NAG	207,208,228,253
3	0.03	4	3w3Nb	RX8	208,210,211,231,233,258
4	0.03	3	4g8Aa	LP5	215,239,240
5	0.03	3	3b2dA	MAN	17,41

**Table 6 animals-11-00032-t006:** Predicted active sites of TLR4 full-length protease.

Rank	Cscore EC	PDB Hit	TM-score	EC Number	Active Site Residues
1	0.162	3h09B	0.381	3.4.21.72	NA
2	0.145	2uv8G	0.309	2.3.1.86	NA
3	0.139	3ecqB	0.312	3.2.1.97	NA
4	0.137	2vkzG	0.306	2.3.1.38; 3.1.2.14	NA
5	0.134	2zxqA	0.311	3.2.1.97	NA

**Table 7 animals-11-00032-t007:** Predicted active sites of truncated TLR4 protein.

Rank	Cscore EC	PDB Hit	TM-Score	EC Number	Active Site Residues
1	0.152	2vkzG	0.332	2.3.1.38; 3.1.2.14	NA
2	0.145	2uv8G	0.333	2.3.1.86	NA
3	0.144	1hn0A	0.329	4.2.2.20	NA
4	0.141	1t3Ta	0.337	6.3.5.3	NA
5	0.138	3ecqB	0.323	3.2.1.97	NA

## Data Availability

Data available in a publicly accessible repository.

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
