# Peer review of "Identification of Yak’s TLR4 Alternative Spliceosomes and Bioinformatic Analysis of TLR4 Protein Structure and Function"

_animals, 2020, doi:10.3390/ani11010032_

Round 1
Reviewer 1 Report
Generic comments:
Please make sure all the tools used within this study are promptly cited by either weblinks or publications.
Please make sure your sentences composed of scientific claims are always back up by literature. Several points of lack of referencing has been commented on in the specific comments section.
The use of a single house keeping gene such as GAPDH across tissues without optimisation could have drastic effect on the relative qPCR results of the study. Could you please explain how and why you decided to only use this 1 HKG? Especially have you tested the unbiased expression of the HKG across the tissues (Geometric mean compassion of your biological replicates in each tissue).
Starting amount of RNA in each tissue sample (i.e. different cell population and RNA content) is a great issue given the uncontrollable nature of first strand cDNA synthesis. Inclusion of RNA extraction metrics and tissue specific metadata as a supplementary table is highly recommended.
Specific comments:
Line 37 – Wrong use of the word “sing”. Please rewrite the sentence.
Line 64 – Perhaps include a suitable reference at the end of this sentence “…crucial immunogenic genes.” e.g. DOI: 10.1177/1753425908095957 or other similar reviews of TLR4 immunochemistry.
Line 87- Please include the version of NCBI’s sequence used along with the corresponding Yak genome.
Lines 98-100 – Please include any ethical approval or animal licence number for the project.
Line 101 – Please change “sacrificed” to “euthanized”.
Line 102 – Please also include the anatomical location the fat tissue was collected from.
Line 110 – Please include extracted RNA quality control metrics (concentration, 280/260 absorbance or RIN)
Line 192 – Please expand the abbreviation in text prior to usage H(Helix) , S(Strand) and C(Coil) .
Figures 6 – 9. Please reframe from including the screenshots as images. The cropping on the right-hand side of the images is removing some of the text related to the figure. Perhaps create a panel of them accordingly with appropriate caption provided for the whole panel or individual figures.
Line 261- Please use MegALign citation for a reference to the tool PMID: 9089607
Line 356 – Please include reference(s)
Author Response
Response to Reviewer 1 Comments
Dear reviewer:
Thank you for your time concerning our manuscript entitled “Identification of Yak’s TLR4 Alternative Spliceosomes and Bioinformatic Analysis of TLR4 Protein Structure and Function” (written by Xingdong Wang et al.), which we wish to be considered for publication in “Animals”. The following is a point-by-point response to your comments.
Generic comments:
Point 1: Please make sure all the tools used within this study are promptly cited by either weblinks or publications.
Response 1: Thank you for your comments. We have revised as suggested and cited references in line 125 Primer Premier 5.0 software [22], in line 152 LOMETS [25], in line 159 BioLip database [27], in line 169 2-ΔΔCT [30], in line 172 SPSS version 21.0 [31], in line 180-181 MegAlign 5.10 [33] and SeqMan [34], and in line 204 Splign [35].
[22] Singh, V.K.; Mangalam, A.K.; Dwivedi, S.; Naik, S. Primer premier: program for design of degenerate primers from a protein sequence. Biotechniques 1998, 24, 318-319.
[25] Sitao, W.; Zhang, Y. LOMETS: a local meta-threading-server for protein structure prediction. Nucleic acids research 2007, 35, 3375-3382.
[27] Yang, J.Y.; Ambrish, R.; Zhang, Y. BioLiP: a semi-manually curated database for biologically relevant ligand–protein interactions. Nuclc Acids Research 2013, 41, 1096-1103.
[30] Schmittgen, T.D. Real-time quantitative PCR. Methods 2001, 25, 383-385.
[31] Qiu, H.; Jin, G.Q.; Jin, R.F.; Zhao, W.K. Analysis of variance of repeated data measured by water maze with SPSS. Journal of Chinese Interative Medicine 2007, 5, 101-105.
[33] Clewley, J.P.; Arnold, C. MEGALIGN. The multiple alignment module of LASERGENE. Methods in Molecular Biology 1997, 70, 119.
[34] Jin, J.Q.; Sun, Y.B. AutoSeqMan: batch assembly of contigs for Sanger sequences. Zoological research 2018, 39, 123-126.
[35] Kapustin, Y.; Souvorov, A.; Tatusova, T.; Lipman, D. Splign: algorithms for computing spliced alignments with identification of paralogs. Biology Direct 2008, 3, 1-13
Point 2: Please make sure your sentences composed of scientific claims are always back up by literature. Several points of lack of referencing has been commented on in the specific comments section.
Response 2: Thank you for your comments. We have revised as suggested and cited references in line 58 1997 [4,5], line 64 genes [11], line 76 site [10] and line 394 functions [49-51].
[4] Du, X.; Poltorak, A.; Silva, M.; Beutler, B.; Analysis of Tlr4-Mediated LPS Signal Transduction in Macrophages by Mutational Modification of the Receptor. Blood Cells, Molecules, and Diseases 1999, 25, 328-338.
[5] Frantz; Stefan; Kobzik; Lester; Kim; Young-Dae; Fukazawa; Ryuji; Medzhitov; Ruslan. Toll4 (TLR4) expression in cardiac myocytes in normal and failing myocardium. Journal of Clinical Investigation 1999, 104, 271-280.
[10] Huang, Z.Y.; Chen, Y.; Li, X.Y.; Zhen, T.; Zhang, Y.; Xu, Q.; Duan, X.J.; Zhao, W.M.; Chen, G.H. Identification and Expression Analysis of Alternative Splicing of TLR4 in Duck. Chinese Journal of Animal and Veterinary ences 2013, 44, 697-702.
[11] Lizundia, R.; Sauter, K.S.; Taylor, G.; Werling, D. Host species-specific usage of the TLR4-LPS receptor complex. Innate Immun 2008, 14, 223-231.
[49] Deng, M.; Scott, M.J.; Loughran, P.; Gibson, G.; Billiar, T.R. Lipopolysaccharide Clearance, Bacterial Clearance, and Systemic Inflammatory Responses Are Regulated by Cell Type–Specific Functions of TLR4 during Sepsis. Journal of Immunology Author Choice 2013, 190, 5152.
[50] Mika, F.H.; Christina, K.; Claudia, M. Toll-Like Receptors in Atherosclerosis. International Journal of Molecular Sciences 2013, 14, 14008-14023.
[51] Oliveira, J.G.D.; Silva, A.E. Polymorphisms of the TLR2 and TLR4 genes are associated with risk of gastric cancer in a Brazilian population. World Journal of Gastroenterology 2012, 18, 1235.
Point 3: The use of a single house keeping gene such as GAPDH across tissues without optimisation could have drastic effect on the relative qPCR results of the study. Could you please explain how and why you decided to only use this 1 HKG? Especially have you tested the unbiased expression of the HKG across the tissues (Geometric mean compassion of your biological replicates in each tissue).
Response 3: Thank you for your comments. The most frequently chosen RGs are so called 'housekeeping genes'-being those required for basic cellular processes. To constitute an ideal RG, a gene must display constitutive, stable expression in all cell types/tissues and treatment regimens under consideration. In vertebrate studies, the most common choice of RGs were ACTB (used in 38% of the studics) and GAPDH (37%) have historically been popular choices. a quantitative review of the literature showed that the average number of reference genes used across all studies was 1.2. Thus, the vast majority of studies continue to use a single gene, with p-actin (ACTB) and/or glyceraldehyde 3-phosphate dehydrogenase GAPDH) being commonly selected in studies of vertebrate gene expression.
In the article [1,2] that Validation of Common Housekeeping Genes as Reference for qPCR Gene Expression Analysis During iPS Reprogramming Process and Optimal Reference Gene Selection for Expression Studies in Human Reticulocytes, it was verified that GAPDH is the most stable internal reference gene, and only GAPDH [3] was used in Bioinformatics and tissue expression of the LYRM1 gene in Guizhou black goat research an internal reference gene.
Based on previous laboratory research, GAPDH has been optimized and stably expressed in yak tissues, so the results of this experiment using only GAPDH are really reliable. The tissue biology repeat in the experiment is 3×3, that is, the same tissue of three cows was used, and each cow was made three parallel.
[1] Yulia, P.; Arno, G.; Shinji, M. Validation of Common Housekeeping Genes as Reference for qPCR Gene Expression Analysis During iPS Reprogramming Process. entific Reports, 2018, 8(1), 8716.
[2] Aggarwal, A.; Jamwal, M .; Viswanathan, G.K. Optimal Reference Gene Selection for Expression Studies in Human Reticulocytes. The Journal of Molecular Diagnostics, 2018, 20( 3), 326-333.
[3] Yang, Y.; Chen, H.L.; Xu, M. Bioinformatics and tissue expression of the LYRM1 gene in Guizhou black goat. Animal Husbandry & Veterinary Medicine,2020, 52(09), 1-6.
Point 4: Starting amount of RNA in each tissue sample (i.e. different cell population and RNA content) is a great issue given the uncontrollable nature of first strand cDNA synthesis. Inclusion of RNA extraction metrics and tissue specific metadata as a supplementary table is highly recommended.
Response 4: Thank you for your comments. We have submitted a Supplementary Form S1 about RNA extraction metrics and tissue specific metadata.
Specific comments:
Point 1: Line 37 – Wrong use of the word “sing”. Please rewrite the sentence.
Response 1: Thank you for your comments. We have made changes in line 37 as suggested to change the sentence that TLR4-X2 sequence and TLR4-X2 predicted sequence sing NCBI were 100% similar change to the sentence that TLR4-X2 sequence was consistent with the predicted sequence of the TLR4 gene in GenBank.
Point 2: Line 64 – Perhaps include a suitable reference at the end of this sentence “…crucial immunogenic genes.” e.g. DOI: 10.1177/1753425908095957 or other similar reviews of TLR4 immunochemistry.
Response 2: Thank you for your comments. We have revised as suggested and cited reference in line 64 genes [11].
[11] Lizundia, R.; Sauter, K.S.; Taylor, G.; Werling, D. Host species-specific usage of the TLR4-LPS receptor complex. Innate Immun 2008, 14, 223-231.
Point 3: Line 87- Please include the version of NCBI’s sequence used along with the corresponding Yak genome.
Response 3: Thank you for your comments. We have made changes in line 87-89 as suggested The sentence that “In this study, the CDS region of the yak's TLR4 gene was amplified using primers, which were designed based on sequences predicted from the yak’s TLR4 gene published in the NCBI “have been changed that “I In this study, the CDS region of the yak's TLR4 gene was amplified using primers, which were designed using yak’s TLR4 gene’ predicted sequences (XM_005891938.1 and XM_014477047.1), obtained using NCBI (https://www.ncbi.nlm.nih.gov/).”
Point 4: Lines 98-100 – Please include any ethical approval or animal licence number for the project.
Response 4: Thank you for your comments. We have revised as suggested in line 102 that Approval No. NKMYD201902: dated: 26 October 2019 has been added.
Point 5: Line 101 – Please change “sacrificed” to “euthanized”.
Response 5: Thank you for your comments. We have changed "sacrificed" to "euthanized" in line 104 according to the amendment.
Point 6: Line 102 – Please also include the anatomical location the fat tissue was collected from.
Response 6: Thank you for your comments. We have changed muscle and fat tissues to back muscles, and the subcutaneous adipose tissue of the back in line 105 according to the amendment.
Point 7: Line 110 – Please include extracted RNA quality control metrics (concentration, 280/260 absorbance or RIN)
Response 7: Thank you for your comments. We have revised as suggested in line 116-121: The RNA concentration and OD260/280 ratio of the samples were determined using NanoDrop 2000 spectrophotometer (ThermoFisher Scientific, Waltham, MA, USA). RNA concentration and OD260/280 ratio of the samples ranged between 500-5000 ng/ml and 1.9-2.1, respectively. The RNA extraction metrics and tissue specific metadata were showed in Table S1. Ribonucleic acid quality was assessed by evaluating the 28S and the 18S rRNA bands on a 1% agarose electrophoretic gel.
Point 8: Line 192 – Please expand the abbreviation in text prior to usage H(Helix) , S(Strand) and C(Coil)
Response 8: Thank you for your comments. We have revised as suggested that expand the abbreviation in text prior to usage H(Helix), S(Strand) and C(Coil) in line 215-217.
Point 9: Figures 6 – 9. Please reframe from including the screenshots as images. The cropping on the right-hand side of the images is removing some of the text related to the figure. Perhaps create a panel of them accordingly with appropriate caption provided for the whole panel or individual figures.
Response 9: Thank you for your comments. We have revised as suggested that Figure 6-9 has been changed, and the main information on the right was placed in Table 4-7 in line 268-278.
Point 10: Line 261- Please use MegALign citation for a reference to the tool PMID: 9089607.
Response 10: Thank you for your comments. We have revised as suggested and cited reference in line 180 MegAlign 5.10 [33].
[33] Clewley, J.P.; Arnold, C. MEGALIGN. The multiple alignment module of LASERGENE. Methods in Molecular Biology 1997, 70, 119.
Point 11: Line 356 – Please include reference(s)
Response 11: Thank you for your comments. We have revised as suggested and cited references in line 394 functions [49-51].
[49] Deng, M.; Scott, M.J.; Loughran, P.; Gibson, G.; Billiar, T.R. Lipopolysaccharide Clearance, Bacterial Clearance, and Systemic Inflammatory Responses Are Regulated by Cell Type–Specific Functions of TLR4 during Sepsis. Journal of Immunology Author Choice 2013, 190, 5152.
[50] Mika, F.H.; Christina, K.; Claudia, M. Toll-Like Receptors in Atherosclerosis. International Journal of Molecular Sciences 2013, 14, 14008-14023.
[51] Oliveira, J.G.D.; Silva, A.E. Polymorphisms of the TLR2 and TLR4 genes are associated with risk of gastric cancer in a Brazilian population. World Journal of Gastroenterology 2012, 18, 1235.

Reviewer 2 Report
The manuscript submitted by Wang et al. deals with the identification of alternative yak’s TLR4 spliceosomes in an attempt to predict the deriving protein structures and functions.
The study is original, well written and the experimental design is clearly presented.
The major criticism concerns the lack of a reference to the document of the ethics committee approval; I believe this document is necessary since 3 animals were sacrificed for the purpose of the study (L101). The statement on lines 98-100 is not sufficient.
In L106-107 authors stated that “Around 3 g of each tissue sample were stored in dry-ice and immediately transported to the laboratory, where it was stored at −80oC until further use”. Why the samples were not placed in preservative solutions before freezing? Even at -80 °C the RNA is susceptible to degradation over time. This point must be clarified.
In paragraph 2.5 Authors must added information about the specific parameters that have been considered in the protein structures preparation (degree of freedom of the bonds, presence of the solvent...).
The resolution of Figures 6, 7, 8 and 9 needs to be improved.
The indication of the bibliographic references in the text must be indicated according to the style of the journal. Replace …[x] with …[x].
Author Response
Response to Reviewer 2 Comments
Dear reviewer:
Thank you for your time concerning our manuscript entitled “Identification of Yak’s TLR4 Alternative Spliceosomes and Bioinformatic Analysis of TLR4 Protein Structure and Function” (written by Xingdong Wang et al.), which we wish to be considered for publication in “Animals”. The following is a point-by-point response to your comments.
Point 1: The major criticism concerns the lack of a reference to the document of the ethics committee approval; I believe this document is necessary since 3 animals were sacrificed for the purpose of the study (L101). The statement on lines 98-100 is not sufficient.
Response 1: Thank you for your comments. We have revised as suggested.
“Animal handling procedures were approved by the Animal Care and Use Committee of the Lanzhou Institute of Husbandry and Pharmaceutical Sciences Chinese Academy of Agricultural Sciences” have been changed to “All experimental procedures and protocols were approved by the Institutional Animal Care and Use Committee of the Lanzhou Institute of Husbandry and Pharmaceutical Science of the Chinese Academy of Agricultural Sciences (Approval No: NKMYD201904; dated: 26 October 2019).” in line 100-102.
Point 2: In L106-107 authors stated that “Around 3 g of each tissue sample were stored in dry-ice and immediately transported to the laboratory, where it was stored at −80 ℃ until further use”. Why the samples were not placed in preservative solutions before freezing? Even at -80 °C the RNA is susceptible to degradation over time. This point must be clarified.
Response 2: Thank you for your comments. The experimental samples were stored in the -80 refrigerator for no more than one week after they were collected. Generally, RNA extraction and reverse transcription into cDNA are performed immediately after the collection, and then stored in the -80 refrigerator in the form of DNA. And the time interval between subsequent experiments will not be very long.
Point 3: In paragraph 2.5 Authors must added information about the specific parameters that have been considered in the protein structures preparation (degree of freedom of the bonds, presence of the solvent...).
Response 3: Thank you for your comments. In paragraph 2.5, we did not prepare protein crystals, but used the protein prediction software I-TASSER to predict the structure and function of the TLR4 full-length protein and TLR4 truncated protein. Starting from the amino acid sequence, I-TASSER first uses LOMETS (a metamthreading algorithm consisting of several separate threaded programs) to identify homologous structure templates or super secondary structure segments from the PDB library, and then constructs the complete structure by reassembling the LOMETS template and the continuous aligned fragment structure excised from the super secondary structure segment, then perfect and optimize the structure model and determine the free energy conformation. In order to deduce the biological function of the target protein, the I-TASSER model is matched with the protein in the BioLip library, and its function is inferred from the BioLip template, including ligand binding site, enzyme classification, etc.
Point 4: The resolution of Figures 6, 7, 8 and 9 needs to be improved.
Response 4: Thank you for your comments. We have revised as suggested. Figure 6-9 has been changed, and the main information on the right is placed in Table 4-7 in line 268-278.
Point 5: The indication of the bibliographic references in the text must be indicated according to the style of the journal. Replace …[x] with …[x].
Response 5: Thank you for your comments. We have revised as suggested. We have re-imported the references in the article using ENDNOTE according to the MDPI format and changed them according to the standards of Animals.

Round 2
Reviewer 1 Report
Thanks for incorporating my recommendations. I have no further comments.